# The Courage to Preach in the Digital Age

Casey T. Sigmon 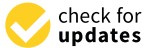

Preaching and Worship, Saint Paul School of Theology, Leawood, KS 66224, USA; casey.sigmon@spst.edu

**Abstract:** This article is an invitation to truly postmodern conversational approaches to preaching beyond a monological modern mass media sermon dressed up in conversational style. This involves proposing a new *who* and *how* for preaching in the digital age, resulting in a New (Media) homiletic. Engaging with Parker J. Palmer's *The Courage to Teach*, this article first explores issues with the traditional top-down model for preaching solidified during the age of mass media technoculture. Next this article names some stumbling blocks to dialogical preaching in a mainline Protestant US context. Then this article explores what difference the implementation of Palmer's Community of Truth model could make for preaching in this digital age. The technoculture of social media allows for and expects preachers to be more conversational in the who and how of sermon delivery, preparation, and feedback.

**Keywords:** dialogical preaching; social media; technoculture; homiletics; Parker Palmer; digital age; new media homiletic



## 1. Introduction

In 2011, Ronald J. Allen revisited his 1991 "Agendae for Homiletics" with "Some Issues for Preaching in the Future." Allen explicitly ponders "how far can the boundary of the notion of preaching extend" and so "prompt us to reconfigure our understandings of the norms for what counts as preaching, expressions of preaching, who can preach, etc.?" (Allen 2011, p. 47). Allen lamented that while homileticians have succeeded in presenting a diversity of sermonic forms in the postmodern era, they have yet to dive deeper into what the emerging "postmodern ethos" could mean for *how* we preach (Allen 2011, p. 47). This article explicitly explores some modern norms that keep the sermon a monologue, that is, one-way communication from the pulpit to the pews, in the hopes that a postmodern practice of dialogical preaching can be given more homiletic appreciation and application in the mainline Protestant church of the United States.

The reason for encouraging a shift in the *how* of preaching, from monological to dialogical, has to do with formation and issues of the empowerment of discerning Christians in our present technoculture. Technoculture is a term that encompasses the reality of how technology and culture reciprocally shape one another. Like all culture, it can be difficult to define because technoculture is water that we swim in day after day. Yet the anxiety, misunderstanding, and rapid changes of our technoculture highlight the fact that we are in the midst of a shift—from the mass media, print technoculture of the modern era and into the social media, network technoculture of the postmodern era.

Homileticians have felt the pressure and ambiguity of this emerging postmodern technoculture concerning the church and its preaching, though they may not use such terminology to define the shifts we are in. From Clyde Reid to Fred Craddock, John McClure to David Lose, Eunjoo Mary Kim to Sarah Travis, Ron Allen to O. Wesley Allen, Jr.—the hunch has been that something needs to change in the *how* of preaching to reach people who are being shaped by postmodernism. Many homiletical solutions have revolved around conversational sermonic hermeneutic and preparatory forms for the monological sermon—subversive postmodern tactics for inhabiting the modern pulpit. The Academy of Homiletics was born in the dusk of modern broadcast technoculture, and the conversation

of the guild reflects, laments, plays with, and subverts this ending. Now the guild must ask what the dawn of postmodern social media technoculture offers concerning the *how* but also the *who* of preaching.

## 2. Prelude: Unfulfilled Expectations for a Priesthood of All Believers

Just over five hundred years ago, a movement ignited that would forever alter the landscape of Christianity. One of the rallying cries in the movement was "the priesthood of all believers", a radical hierarchical shift in church structure in 1517. Imagine the radicalism of "the priesthood of all believers" as it beckoned the church to consider the notion that there is no spiritual divide between the baptized (of course, to varying degrees of equity in the many streams of reform). Instead, in the sight of God, said the movement, all are considered priests connected to God.

Martin Luther is often credited for the phrase, "priesthood of all believers", though we do not have one piece of writing to cite as the source of this phrase. Luther's idea of the universal priesthood emerged from his reading of 1 Peter 2:9, "But you are a chosen race, a royal priesthood, a holy nation, God's own people, in order that you may proclaim the mighty acts of him who called you out of darkness into his marvelous light" (NRSV).

Although Luther offered this challenge to the church, neither he nor most of the Protestant mainline movement have fully lived into a vision that bridges the divide between clergy and laity. Luther himself restructured the church in such a way that the Word of God would become the center of power, meaning in practice that the one ordained to preach would ultimately be the emissary of the living word. The *who* of preaching revolved around the minister/priest and the *how* in the architecture of pulpit to pews. Reformed Protestant worship space would have one ornate pulpit and settled beneath it, rows upon rows of pews—an architectural snapshot of the evolving technoculture born of the print age with those even lines broken up by the binding of the center aisle. As far as preaching was concerned for most of the movement the priesthood of all believers did not erase liturgical roles, notably preaching—meaning that the one who benefitted the most from daily wrestling with the scriptures and the public theologizing from that wrestling would still be the priest/pastor. If the preacher worked at clearly articulating the message, the assumption was that the people in the pews would successfully receive the transmission and apply it to their life.

Luther's discomfort with a preaching-hood of all believers is revealed, especially in his disputes with the Radical Reformation and its Anabaptist leaders. Here is Luther on the anarchy of Anabaptist preaching:

> First one layman would stand up and disagree with the preacher, then another would shout down the first. Soon all the men would be shouting among themselves, and even the women would demand a say and tell the man to be silent. The services would sound more like the annual fair than a church (Grieser 1997, p. 535).

Luther opined that in the communal preaching of the Anabaptists, no evidence of the "simple believer delivering the sermon itself" was to be found (Hartshorn 2006, p. 37).

Luther tried in vain to bring the Anabaptists back into the state churches. When he inquired as to why they would not worship with the Reformed Church, the Anabaptist reply was that if there is no space for "interactive response" or "mutual admonition" in and around preaching, then according to the Anabaptists, the church was not Christian order as taught by Paul in 1 Corinthians 14:26b-33a (Hartshorn 2006, p. 37).

> When you come together, each one has a hymn, a lesson, a revelation, a tongue, or an interpretation. Let all things be done for building up. If anyone speaks in a tongue, let there be only two or at most three, and each in turn; and let one interpret. But if there is no one to interpret, let them be silent in church and speak to themselves and to God. Let two or three prophets speak, and let the others weigh what is said. If a revelation is made to someone else sitting nearby, let the first person be silent. For you can all prophesy one by one, so that all may learn

and all be encouraged. And the spirits of prophets are subject to the prophets, for God is a God not of disorder but of peace.

The Anabaptist thread of communal preaching practice is entangled in the last 500 years of preaching history, a prophetic whisper of the Reformation's original rallying cry. In practice, the dominant form of preaching is the form of a pulpit for one speaking to rows of pews. As a result, preaching has contributed to the spiritual divide between clergy and laypeople. The architecture of our worship spaces reinforces this divide, be it in a neo-gothic Episcopal Church with its pulpit or a non-denominational megachurch with its music stand on center stage.

### 2.1. The Mass Mediated Sermon: The North American Protestant Who and How of Preaching

As Christianity in the United States cross-pollinated in the modern era with the cult of personality, the rising star system of Hollywood, and consumerism, preacher celebrities found fertile ground to grow large audiences—both in the mega-sanctuary and through evolving broadcast mediums that brought the preacher into the homes and onto the big screens of Christians everywhere.

Mass media is a term coined in the early 1940s by Harold Lasswell. However, it has existed since the creation of the printing press (Hardt 2004, p. 15). Lasswell used the term in the context of government and related it to propagandistic activities on both sides of World War II. Mass media is the technical use of any media in order to (re)produce knowledge and information efficiently—reducing dialogical relations in order to amplify the mediated message at the expense of conversation (Hardt 2004, p. 14). Mass media is by nature a one-way message system privileging the distributor of the message. Distribution is not in the hands of the public. The flow of communication is top-down. The public is formed for receptivity and consumption of the message coming through the mass media pipeline.

Pulpit preaching as many American Protestants experience it, especially white congregations, overlaps with the communication systems of mass media and broadcast media. The preacher is the broadcast anchor, the official reporter of truth. The laity is the audience, the viewership formed over time to receive the truth from the expert and reproduce the facts to others in their orbit.

Congregations in the post-pandemic era have a particular decision to make—to continue to simply transmit the livestream, one-way worship service and sermon that is a product of the modern mass media era or to innovate and explore means to allow for a participatory postmodern homiletic to emerge.

The digital age, especially the social media technoculture, disrupts the tradition as it amplifies the fruit of our centuries of ecclesiological formation based on a top-down communications model. Our technological advances have simultaneously introduced the technoculture of Web 3.0—a radically vertical and non-boundaried lifeworld for public conversations and connection—and amplified the means for mass mediation. Not all online preaching models a new *how* and *who* into practice. Satellite preaching, livestreaming, and the phenomenon of ordering DVDs of great preachers to play in the church reflect the one-way amplification of the preacher's voice.

I perceive one of the greatest divides in the United States to be between an older generation accustomed to, and satisfied by, mass media technoculture (ecclesiologically reflected in traditional churches, as well as megachurch and satellite church ecclesiologies) and a younger generation suspicious of mass media who are seeking instead greater collaboration, conversation, and participation. Indeed, it seems the new media of this age (Twitter, smartphones, Facebook Live, etc.) are luring the church into novel ways of breaking down the divide between professional Christians and lay Christians, which is irksome to many established Christians who are taught not to talk while the preacher is talking.

In order to reflect on the patterned who and how of preaching out of the modern mass media technoculture and diagnose what may need to be overcome in the preacher and in

the people to shift into a new homiletic for the social media technoculture, let us explore Parker J. Palmer and his book *The Courage to Teach*.

### 2.2. Mythical Objectivism: A Shape for Preaching from the Mass Media Age

> Like most professionals, I was taught to occupy space, not open it: after all, we are the ones who know, so we have an obligation to tell others about it (Palmer 2007, p. 135)!

Certainly, a new *who* and *how* proposed for a communal paradigm for preaching is, well, scary. Why? What if no one speaks up? Or what if too many want to talk? Will we be done in an hour? What if someone sputters off utter nonsense or bad theology? Will we be led astray? Have you seen what my congregants are sharing on social media feeds? Who will oversee the preaching and correct heretical comments? My invitation is for preachers to replace fear of what *might* happen in conversational preaching with the courage to step into the mysterious possibility of what *could* happen over time to the benefit of the church. At the same time, those who have been preached *to* for so long will need to find the courage to speak up in church on faith matters after centuries of distancing them from the Bible and proclamation through the liturgical choreography of the modern mass media technoculture.

A potential visual prompt for preachers to reflect on the ecclesiological formation American Protestant Word Centered mass media worship has on the laity (especially when laity are not participating in studies and small groups outside of the worship hour) comes from Parker J. Palmer's classic *The Courage to Teach*. In it, Palmer spends significant time speaking to the malformation that has taken place in learning and teaching, hinted at in the quote above. Teachers were taught to take up space in the classroom with their knowledge and expertise and to deliver that information to the students who otherwise would not know what to know or how to know. In this next section, we will focus on Palmer's observations about truth: what it is and from where it emerges. In other words, these models from Palmer speak to the how and who of teaching, and very well can enrich our homiletical reflection and practice.

In what Palmer calls "an earlier, more naïve era when people were confident, they could know the truth", educational models were formulated under the paradigm of "mythical objectivism" (Palmer 2007, p. 102). Figure 1 illustrates the paradigm. Palmer calls this model "mythical", yet it remained the dominant model of truth-knowing in education and had real, material impacts on the formation of young people in the modern era of education (Palmer 2007, p. 102). To paraphrase the model, objects of knowledge reside somewhere out there, pristine in physical or conceptual space, described by facts in a given field that only an expert can pass on to others lower down on the chain of learning. "The truth is out there", as some might say. Truth is not inside of the expert, because subjectivity is suspicious and less than rigorous, but the expert knows how to report truth without clouding up the facts with their bias. Furthermore then, truth is not inside of the amateurs. Experts are people trained to know objects in their pristine form without allowing subjectivity to get in the way. Training occurs in special schools where one is formed to be a trustworthy (read subjectless) bearer of the pure object.

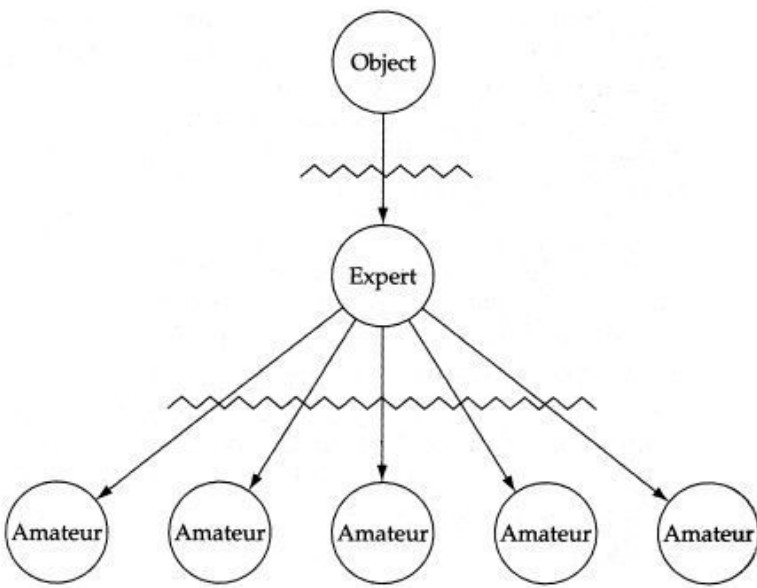

**Figure 1.** Parker Palmer on "Mythical Objectivism".

Palmer speaks at length about the consequences of this "mythical objectivism" on teachers and learners in the West. Palmer argues that people cannot truly learn this way. At this juncture, let us pause to consider what this myth has to do with preachers in the West who indeed were seminary trained in this model.

This assumption that truth is acquired through specialist training leaves us with a church full of amateurs. The *who* of preaching involves three parties:

1. God (Holy Three-in-One, the object of our preaching, revealed through the Word).
2. The Preacher/Expert/Master of Divinity (the one entrusted with bringing God's Truth down to the pews with expert knowledge of scripture, theology, tradition, and history without cloudy subjective bias getting in the way).
3. Laity/Amateurs (without rigorous training and with subjective bias, so the system renders them dependent on the expert for objective or pure knowledge of God).

In this framework, the *how* of preaching revolves around accurate rhetorical application to ensure the laity/amateurs rightly receive the preacher/expert's transmission of the facts about and from God.

Modern homiletical theory, as stated earlier, emerges from this mass media, top-down technocultural form. The guild's research, writing, and teaching has responded to the problems and possibilities of this paradigm for over six decades. Homiletic theories are offered to the expert preachers to equip them to bring their expert engagement with the word of God into contact with the laity/amateurs. Some homileticians, such as David Buttrick in Homiletic, explicitly discourage the muddying of preaching with the preacher's explicit subjectivity (as if he could ever allow the wisdom gathered from study of scripture to not be entangled with his personhood, bias, and ways of knowing).

Feminist homileticians (such as Christine Smith, Lucy Atkinson Rose, and L. Susan Bond) and Womanist homileticians (such as Teresa Fry Brown, Gennifer B. Brooks, Valerie Bridgeman, Melva Sampson, and Lisa L. Thompson) resist the myth through highlighting the strengths of the preachers' subjectivity and awareness of subjectivities when preaching. Yet, aside from a much needed who in the pulpit that is the non-White, non-Male, non-binary preacher/expert, the who and how remain unchanged.

*2.3. Preaching and the Monological Illusion: Dusk for the Mass Media Age*

In 1963, Clyde H. Reid wrote a concise yet compelling argument against preaching's efficacy in the top-down, mass media, broadcast form. In "Preaching and the Nature of Communication", Reid cites new discoveries in communications studies that show how an

absence of dialogue leads to a communication breakdown in the intended recipient of a message. Reid says, "Until about 1950, communications researchers thought of communication chiefly as a simple, one-way process" (Reid 1963, p. 41). Notably, researchers such as Melvin L. DeFleur and Otto N. Larsen link problematic issues in classical sociological and psychological theories and connect them to a critique of mass communication. Reid adapts these insights and applies them to preaching, saying, "A basic assumption of this paper is that preaching is an attempt to communicate, and if preaching is communication, then it must be subject to the laws of human communication" (Reid 1963, p. 41). Bringing in Reuel L. Howe and the "monological illusion", Reid's conclusion is firm: "The absence of dialogue in the preaching situation may be the key to understanding the failure of preaching to achieve the results in changed lives" (Reid 1963, p. 43).

Reid does not stop at a critique of the sermon monologue. He goes on to critique the church order that emerges from mass media technoculture's impact on preaching: a paid preaching expert upon whom the lay folk rely upon to "perform" the "bulk of the church's ministry" (Reid 1963, p. 8). Additionally, Reid laments, "By turning the preaching ministry over to a paid professional, we are also giving him a job too big for one man alone" (Reid 1963, p. 49).

Holding Palmer's critique of mythical objectivism up with the critique of the monological illusion in preaching, we wonder then how much our congregants learn about God from the traditional, monological sermon. If, according to Palmer, students will not truly learn under this paradigm, nor Reid through monologues, what is the future of homiletics? Let us contemplate how the top-down, mass media who and how of preaching can reach a culture increasingly suspicious of institutional authority and shaped by social media technoculture.

Homileticians are making a shift in conversation within the guild, as evidenced in part by this special issue of *Religions*. In 2013's *Preaching at the Crossroads: How the World—and Our Preaching—Is Changing*, David Lose claims that we are at a crossroads in preaching, requiring us to question the fundamental pattern of a practice we have come to rely on for centuries. Lose notes how the Internet, specifically Web 2.0 at the time of his research and writing, reveals a preference for interactive ways of coming to know and constructing identity rather than receiving information and identities from authorities. Perhaps, Lose suggests, we could learn from the shift toward open-source Web programs requiring user interaction in our preaching. This would require a shift from what Lose sees as the predominant homiletic—a performative homiletic—to a participatory one.

Whereas the performative homiletic views the preacher as the sole and chief interpreter of Scripture and Christian identity, (Lose 2013, p. 105) the participatory homiletic sees the preacher as a creator of space for the congregation to become fluent interpreters of the Christian faith (Lose 2013, p. 107). Appealing to shifts in technology over the past two decades from static and consumer-driven postures of Web 1.0 to the emerging interactive platforms and postures of Web 2.0, Lose imagines the sermon as a "transport mechanism, the ether through which interactivity [between God's word and God's people] happens" (Lose 2013, p. 109). Space is created in the sermon for the congregation to interact and participate rather than merely watching the performance of the proclamation.

Ultimately, Lose suggests practicing this participatory homiletic relies on practices lifted up by McClure and Rose in their roundtable and round pulpit efforts, so he does not go beyond metaphorical proposals for the practice of communal preaching. The place for preaching is still Sunday, from a pulpit to a pew. The preacher is called to better connect with her congregation in the real world by visiting their place of work. The preacher has guided conversational sermons with congregants in the pulpit. The preacher has a small group come together to reflect on the upcoming text for the week, and their voices make it into the sermon. However, Lose cannot imagine for the reader a form of preaching that involves, in real-time, a community of proclamation.

### 3. Communities of Truth: A Shape for Preaching from the Digital Age

> We have systemic problems, which is to say the problems facing the church and all of humanity are a series of interconnected, interanimating, and interdependent problems (Friesen 2009, p. 22).

Lose encourages the preacher to create space in the sermon for interaction and participation. The *how* for this space-making involves practices leading up to and after the sermon but not hosting dialogue in real time. The *who* remains the preacher. Parker Palmer's alternative model for impactful teaching and learning also speaks to creating space in the classroom for interaction and participation between and amid a community of knowers. Palmer's proposal to pedagogy is that truth is not a pure object waiting to be found by a few experts. Rather truth comes about by awareness of the web of communal relationships in which we are permanently embedded. According to Palmer, "truth is an eternal conversation about things that matter, conducted with passion and discipline" (Palmer 2007, p. 106). Truth, then, is never found and concluded. Instead, it always emerges in the ebb and flow of the web of community.

Teaching this model of teaching/learning is more complex than the model of mythical objectivism. Rather than teachers focusing energy exclusively on banking all there is to know about a subject in their stores alongside techniques for transferring that information into students, they learn capacities for holding tension, hosting dialogue, and holding space for truth to emerge in relationship amongst a community of knowers. Thus, Palmer calls on the system of educating educators to put forward more person-centered curricula, advocating for the inner life of the teacher to be the starting place for teaching to occur. A key obstacle Palmer names is fear. The lectern in the mythical objectivism paradigm served as a barrier to encounter with others who could invoke fear within the teacher: fear of conflict, fear of diversity, fear of the very real possibility of transformation when another's experience challenges our own.

The pulpit can also serve as a hiding place for the preacher and their fears, though this is not universally or necessarily so. I have seen pulpits encourage preachers to proclaim without fear of retaliation from, rejection of, or apathy toward the gospel. Nonetheless, one of the barriers I often hear from students when I bring up the practice of conversational preaching is fear of what might emerge, conflict, tension, falsehoods, and heresy. We echo the frustration of Luther over Anabaptist preaching, "The services would sound more like the annual fair than a church" (Grieser 1997, p. 535). Implied here is the notion that church is the space where the Word of God, spoken through the mouth of the preacher, is proclaimed. The congregation's role is to be still and know. These cases against conversational preaching have something to do with the self-fulfilling prophesy of mythical objectivism. The laity are amateurs. God could not possibly speak through them without us (preacher/experts).

O. Wesley Allen, Jr. is a homiletician who strives to get beyond mythical objectivism by highlighting ecclesiology and indeed challenging the notion that preaching should be hierarchically defined by the sermon from the pulpit. In *The Homiletic of All Believers: A Conversational Approach*, Allen articulates a matrix for preaching, decentering the preaching event as the ultimate proclamatory act. Allen's matrix is a powerful vision for a sermon that is without end and woven between many partners in a consistent "give and take" of reciprocal communication (Allen 2005, p. 40).

Allen remains tethered to the sermon as performance of one, saying "this shift in the theology of preaching that I am proposing does not marginalize or devalue the pulpit . . . it places the pulpit in its proper place" (Allen 2005, p. 40). Even though he seeks to demote the rank of the pulpit in the matrix of theological conversation as on par with other moments, the *who* and *how* of preaching remain the same. However, a homiletic shaped by social media technoculture seeks openness to conversational preaching in ways Allen seems unwilling to venture.

The most pointed break away from conventional homiletics in what is being proposed may not be the *how* of preaching, but the expansion of *who* can preach. Homiletics so far has been a field of study embedded in the modern Western mass media technoculture.

Embedded in this technoculture, homileticians have struggled, whether they knew it or not, to break away from the oppressive power lines of the top-down mythical objectivism. God is behind the expert/preacher who enters the holy of holies to study the Word and bring its truth to the amateur/laity with rhetorical precision and without bias. This tethering has resulted in stasis around the who and how of preaching, which has waylaid homiletic creativity and innovation beyond the top-down form seen in Figure 1. It has also allowed only certain authorities, namely the preacher, to be designated with a correct grasp of the Scriptures, leaving the priesthood of all believers out of touch.

### 3.1. Touch as Biblical, Contextual, and Technocultural Hermeneutic: Challenging the Grasp of 'Authorities'

Interpretation of all sources for knowledge and practice, be it scripture, tradition, experience, or reason, is both affirmative and critical in a homiletic shaped by new media. Being out of touch with our sacred text and traditioned dogmas is not an option. Nor is passive grasping of them as truths that are never-changing and never-challenged by changes in history and culture. To be in conversation as a community of truth requires a posture of touch. Through touch, the community seek*s* connection with the past and present, allowing the authorizing force of life lived today to speak to the authority of one*s* who lived before.

The preacher in a New Media homiletic seeks to build trust as a hub of connection to the Living Word, revealed through scripture. Authority is relational in nature, not attached to the title "Reverend" alone. It is also kenotic, requiring the refusal to assume power-over. Authority to host a gathering around the Word of God is exercised through the cultivation of creativity and pastoral sensitivity. The preacher does not persuade and manipulate or broadcast and sell the gospel. Rather they seek to make connections in life between tragedy and hope, mundane and sacred.

In sum, the aim of preaching in a New Media homiletic is not to grasp the audience with our message of truth. Rather, truth is understood in ways like Palmer's pedagogical approach, as "an eternal conversation about things that matter, conducted with passion and discipline." The things that matter include scripture and lived experience of all knowers. This emerging model, like Palmer's subject-oriented method of teaching, blurs the clear boundary between audience and speaker. It disrupts the binary by placing the Living Word of God at the center of the sermon event. Though technologies make it possible for preachers to broadcast gospel and amplify our presence in ways unimagined before—a posture that is not rooted in new media technoculture yet open to engage and inform it—reminds us that such a goal is not fitting to the work of allowing gospel to emerge through dialogue and mutuality.

### 3.2. Moving from Fear into Courage in the Dawning New (Media) Homiletic

There are many barriers to the formation of a community for shared proclamation. As mentioned earlier, most of them are rooted in fear. Fear of saying the wrong thing if you have been formed in church and school to see yourself as an amateur Christian. Fear of hearing from someone who may challenge what you believe or read in a book about theology. Fear of different perspectives moving through conflict. The *how* of this preaching method is still important. It will center on methods for creating space in which the truth can emerge within community, through fear, tension, and conflict. If power is not checked and trust is not nurtured through the liturgy, the truth will not emerge, or what arrives will be compromised. Rather than establishing more rules, it will be better for the pastor to nurture principles that the community can agree to. This practice is seen often in social media technoculture around group or page rules that a group of moderators posts and enforces in order to handle the crowd (Gupta 2020).

The pastor will also take responsibility for choosing and framing the subject delicately and intentionally so that the community can gather around "the thing" and explore it together. It is a slow movement, very much counter to the sort of pedagogy that emerges

from the former model, wherein "we keep dumping truckloads of sand on our students, blinding them to the whole, instead of lifting up a grain so they can learn to see for themselves" (Palmer 2007, p. 125).

Perhaps the reader is questioning whether or not this New Media homiletic is truly conversational if the pastor is setting the theme and hosting the dialogue. Palmer addresses this tension as being between two "A's:" Authoritarianism and Anarchy. If one swings too far over toward Figure 1 and the top-down structure of teaching, then truth is rendered static wherein ideas are inert and lifeless. For the classroom, Palmer argues, this dictatorship will not lead to learning. On the other hand, Palmer is not inviting anarchy into the classroom. That is, learning does not happen with an anything goes, hands-off approach. In a classroom setting designed like Figure 2 without a teacher, the dialogue can easily slip into a *Me! Me! Me!* subjectivism with its own threats to thwart connection (Palmer 2007, p. 105). The skilled teacher in Palmer's model collaborates with the knowers to co-create the space of learning through a series of six "both/ands", such as creating a space that is bounded and open (with a theme/subject in focus) and honoring little individual stories as well as the larger Story of a subject or field (Palmer 2007, pp. 76–77).

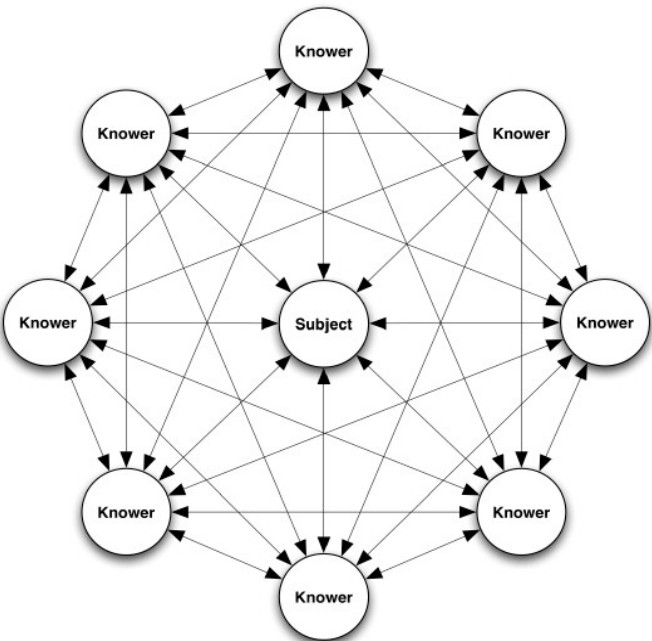

**Figure 2.** Parker Palmer's Subject-Centered Pedagogy-A Community of Truth.

In the case of a pastor, they must have the courage to navigate gathering a Word-centered community between the poles of authoritarianism and anarchy. They become a hub of connection that generates revelation between knowers, text, and tradition. All have equal presence with the Living Spirit of God, as close to each as their own breath. The pastor helps to remind a body long held at pulpit's length from God that they are indeed able to be in touch with God.

### 3.3. Rechoreographing the Priesthood of All Believers in the New (Media) Homiletic

There are rewards for this sort of shift toward shared proclamation. For one, it nourishes communities of proclamation in which everyone takes responsibility for theologizing or talking about God from what they know from their inner life in dialogue with a community. Communications studies including those cited earlier in the Reid article prove that we learn best by being able to respond immediately to information that is offered rather than sitting and passively receiving it. Knowledge of God and the world is shared and spread throughout the community. Congregations increase participation. Every day Christians find their voice.

Moreover, there are rewards for the pastor who witnesses this communal event. They hear the hearts of their congregations. They hear the hopes and fears rather than sitting in her office and imagining what those may be. The leader receives guidance on how best to lead the congregation and where to go. The divide between preacher and people crumbles as lines of communication are now open both ways. Furthermore, the possibilities for proclamation, spoken and embodied, are endless, both in the conventional sanctuary and in online or hybrid spaces where the church gathers for worship.

From Palmer's book *The Active Life: A Spirituality of Work, Creativity, and Caring* we also get a glimpse of the benefits for the congregation and wider church as they are liturgically choreographed toward response-ability and agency in their spiritual lives. It offers a new vision of spirituality that is grounded in the everyday activities of our lives, rather than in traditional religious practices, notably attending worship for one hour each week in a largely passive/receptive posture. Palmer emphasizes the importance of listening to our inner voices, cultivating our creativity, and engaging in meaningful work that contributes to the wellbeing of others. He offers practical advice and exercises for readers to deepen their own spiritual practices and live more fulfilling lives of contemplation and action in community (Palmer 1999).

In a 2022 essay, I wrote specifically about a framework for cultivating spiritual postures informed by the Christian tradition for social media technoculture (Sigmon 2022). Social media literacy and the practice of spiritual disciplines in response to technoculture are part of a New Media homiletic.

Consider Figure 2 as a model for the worship space where and when a sermon takes place. I have preached in spaces where the pulpit is centered within the gathering rather than set apart from the people. This shift in the placement of the pulpit is a step in the conversational direction, but it is not an example of conversational preaching, nor the community of truth described by Palmer. I, the preacher, am not the subject. Rather, God is. God is at the center. Our job as preachers becomes the skillful work of framing the encounter of a community of knowers. We gather them around the subject at hand on a given Sunday so that they will not rely on me to perform the gospel. Rechoreographing them in this New (Media) homiletic, the priesthood of all believers (knowers) will learn to encounter and proclaim the good news for themselves and to appreciate, rather than fear and flee, difference.

Now the sermon event becomes a partner to the work of the people, rather than a hindrance to participation. The pastor does not serve as one without authority. Rather, they serve as one with a different sort of authority that is not a reflection of top-down models. It is a relational authority instead, living and not static. In practice, the pastor designs the sermon event by concise and open-ended framing of a theme or idea from scripture. Beforehand, along with study and dialogue in preparation for the sermon series and individual sermons, the pastor also cultivates thought-provoking questions, ones that are open-ended and invite conversation. In many ways, an emerging New Media homiletic looks an awful lot like the preaching we see Jesus doing in the gospels. Framing an event with questions that get at the heart of the matter about who God is and what the Kingdom of God is like.

The key to Palmer's proposal, and so to mine, is that the preacher/teacher is a knower on the periphery of the subject along with fellow knowers and children of God. Can we rearrange the liturgical homiletic furniture in such a dramatic fashion? We can. We have. With courage, creativity, and discernment.

### 3.4. TweetChat Preaching: A Brief Example for Homiletics in a Social Media Technoculture

Digital spaces have already nurtured preaching that reflects conversational preaching in a New (Media) homiletic. I have had many opportunities to participate in @TheSlateProject's weekly TweetChat known as #SlateSpeak. Every Thursday, for one hour starting at 9 pm EST, communal preaching in the style of lament, confession, conversation, question-

ing, and imagination takes place among a network of lay and ordained pastors seeking renewal for the church and accountability in their walks with God.

The way the TweetChat operates is a bounded time and space for a guided conversation around a particular theme, question, event, or text. Each week, a different moderator selects the centering subject. They will introduce themselves leading up to the start time and share ahead of time what the topic may be. As participants enter the chat on Thursday evening, they will often offer short introductions, especially those new to the gathering. Then the moderator will offer an introductory prayer tweet to open the preaching event.

The moderator likely has a series of 5–6 guiding questions in mind before the start of the chat, each building on anticipated comments from the previous question. They will throw out the first question and wait as participants begin to respond. Strong moderators model dialogue by commenting on the comments of others, tagging participants in comments, and connecting threads to one another's tweets. Once the first question seems to reach an end of energy, another question is introduced, and so on, until the hour is through. At that point, the moderator may summarize, call for prayer requests, and close out the event with a benediction. Some may stay and continue the TweetChat but most leave and return the following week.

TweetChats only scratch the surface of novel ways of communal preaching made possible with social media. There are glimpses of this change in non-digital spaces as well. They are found in non-traditional spaces where homiletical imaginations can untether from the monological mold. Dinner churches gathering around tables and pub churches gathering at the bar or around darts are proclaiming in conversational ways. The spatial organization and implementation, without clear boundaries between the preacher and the people, reflects a truly conversational preaching style—mutual listening, open questions, organic, subject-centered flow. The preacher as moderator/host holds space for truths to emerge but no longer takes up the space as an expert of one.

## 4. Conclusions

We are amid rapid technocultural change in what folks tentatively label the digital age. Even before March 2020 and its push to move worship, including preaching, online, this change has reframed how we relate to one another, connect, and come to know ourselves and our world. We live in a global network with ever-present and evolving tools that have lured hundreds of millions of people into a daily reality known as X-reality, with relationships, connections, and conversation informing and forming us through a constant flow blurring the lines between "virtual" and "conventional" reality in the palm of our hands or the band on our wrist. The novelty of this moment offers homiletics new ways to preach—a new *how* and *who* for preaching—for those willing to embrace the new possibilities within technoculture for reimagining the event, spaces, and media in which we preach. The method for how we preach the Good News of Jesus Christ could be done in ways that better reflect the priesthood of all believers that is in the DNA of the Reformed tradition as it comes into conversation with the postmodern new media communication practices of Web 3.0 technoculture.

These examples strive toward greater inclusivity and activity from preaching partners to strengthen the wisdom and discernment of the whole Body of Christ: a novel *who* and *how* for preaching as it leaves the modern age. The new preaching practice involves risk. There is risk on the part of the preacher, who no longer controls and filters on their own *the* message, point, application, focus, and function of the sermon event. There is also risk on the part of the community, who has been shaped in church culture to sit and listen to the paid expert Christian in their midst rather than talk back and with the preacher.

Yet one wonders what the formational impact could be within this technoculture of echo chambers and algorithmic editing if more preaching was indeed conversational. Could we rewire our relational pathways out of polarization and into complexity? Could we rehumanize the others on the "other side" of "us" and learn new ways of being, voting, and organizing out of these conversations? What will happen if preachers spend more time

listening and less time taking up space? What will happen when laity spends more time thinking through a response to the Word rather than passively sitting back for 15–45 min during the sermon?

The glimpse of a model and practice for conversational preaching mentioned here still relies upon the focused study and preparation of the ordained preacher in tension with the capacity to be an improviser and host to the other partners in the preaching event. Plus the model will put more responsibility on the laity to participate as fellow knowers in an ongoing encounter with God's Living Word. In the comedic sense, Improv relies upon the mantra of "Yes, and" meaning the capacity of fellow performers to listen to one another and to respond in ways that do not pull the rug out from underneath the proposal set in motion in the beginning of a scene. Successful improvisation promotes collaboration and the affirmation of the world being hosted through collective imagination. One may even be surprised to learn that one of the keys to successful improvisation is generosity. Legends of North American improvisation Del Close and Charna Halpern put it this way: "A truly funny scene is not the result of someone trying to steal laughs at the expense of his partner, but of generosity—of trying to make the other person (and his ideas) look as good as possible" (Halpern and Close 1994, p. 16). The result is a story or scene greater than the sum of its parts, which emerges from the synergy of the collaborating whole. If any member decides to go against the grain to shift the flow radically, the synergy falls apart.

So may it be with our preaching in a proclamation of all believers. The truth that we may find together will be a truth that finds, moves, and transforms us and empowers us to be the Body of Christ in the complexity of this digital age.

**Funding:** This research received no external funding.

**Data Availability Statement:** Not applicable.

**Acknowledgments:** Parts of this article are adapted from Casey T. Sigmon, "Engaging the Gadfly: A Process Homilecclesiology for a Digital Age", PhD diss. Vanderbilt University, 2017.

**Conflicts of Interest:** The author declares no conflict of interest.

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
