# Peer review of "The Courage to Preach in the Digital Age"

_religions, doi:10.3390/rel14040551_

Round 1

Reviewer 1 Report

The paper does a nice job laying out the theological and homiletical argument toward a more democratized sermon that draws on Web 3.0 technologies. The paper would be strengthened if the author attended to more concrete examples of digital proclamation and scholars making advances on this front. The paper would also benefit from drawing explicit connections between how the author situates Luther's priesthood of all believers with the author's stated inclination toward process theology.

A more in depth engagement with Reid's 1963 article would be helpful. The use of David Lose's Preaching at the Crossroads is a bit dated. Some better conversation partners for this project might include Melva Sampson ("Going Live"), John McClure (Mashup Religion), Jacob Myers (Curating Church), Rob O'Lynn (Digital Jazz), and Dominique Robinson (iHomiletic). Also, I recommend that the author engage Leah Schade's Preaching in the Purple Zone, which advances a similar argument for a democratized homiletic in non-digital spaces.

Author Response

Gratitude to the reviewer for these strong suggestions. I am currently writing another full book manuscript for this research project, so those authors appear in the broader project and are not in this article which will not be in the book and is an adaptation of an earlier unpublished essay.

Attached is a revised draft with another round of copyedits.

I will note that Lose's book is two years younger than McClure's Mashup

Reviewer 2 Report

This is an excellent and timely essay!

The main question being addressed is what preaching will look like in coming generations, especially how congregants can be included in the preaching moment.

I consider the topic original and relevant to emerging conversations in homiletic and ecclesiastical concerns. I think it addresses the gap in homiletics related to the role of preaching in the process of discipleship.

This is a unique contribution that broaches a needed conversation. It provides both language and methodology for engaging in this conversation.

As presented, I think the author's methodology is sound.

I believe the conclusions are consistent with the evidence and arguments presented and the references are appropriate.

Author Response

Thank you for these notes of encouragement.

Reviewer 3 Report

1. Corrections to phrasing required at lines 125, 128-29; to spelling or formatting lines 151, 213, 227, 242 (possibly elsewhere).

2. The article avoids any mention of Scripture, which has historically served as the focus of interpretation/conversation within Reformational ecclesiology (which appears to be the writer's context). Accordingly, some discussion is required of how, according to this model, multilateral homiletical imagination is (or is not) to be normed by Scriptural texts, if indeed its focus is on "the Good News of Jesus Christ" (line 437).

3. Similarly (but from a Catholic/Orthodox perspective), this model of a mutually informative and multilateral homiletic envisages only those immediately present as conversation partners, with no mention of voices from the past or from elsewhere in the (non-Western) world. Might a truly egalitarian interchange attend to these voices also?

4. Despite its radical democratization, the proposed "participatory homiletic" still endorses a view of "the preacher as a creator of space for the congregation to become... interpreters of the Christian space" (lines 262-63, citing Lose) and the pastor as the one who will "take responsibility for choosing the subject" for community discussion (lines 354-55). Some explanation (whether theological, ecclesiological, or organizational) is required for retaining such privileges if all participants truly have an equal voice.

5. The article cites Parker Palmer primarily in his critique of mythical objectivism, and only more briefly with respect to subject-centered pedagogy. This imbalance could be remedied by citation of Palmer's extensive discussions of spirituality, for example his concept of expressive action (in The Active Life: A Spirituality of Work, Creativity, and Caring), which seems especially germane to this argument.

Author Response

Gratitude to the Reviewer for this careful read.

Attached is an updated version of the article with the following changes/additions:

  1. Corrections to phrasing at lines 125, 128-29; spelling or formatting lines 151, 213, 227, 242 (and other spots as discovered).
  2. An additional section on Scripture and how it can be approached in the New Media homiletic. More extensive writing on this subject will be included in the larger research project/book.
  3. In the larger project, Eunjoo Mary Kim's Transcontextual Hermeneutic is engaged. But given the confines of this article, not integrated here.
  4. Clarification on the needed role of a pastoral guide in the conversational homiletic.
  5. Additional paragraph on spirituality in Palmer and a reference to a previously published article I wrote on the need to cultivate lay spirituality in the digital age for communal belonging.
